# Effect of a Six Week In-Season Training Program on Wrestling-Specific Competitive Performance

**DOI:** 10.3390/ijerph19159325

**Published:** 2022-07-30

**Authors:** Lucciano Francino, Bayron Villarroel, Pablo Valdés-Badilla, Rodrigo Ramirez-Campillo, Eduardo Báez-San Martín, Alex Ojeda-Aravena, Esteban Aedo-Muñoz, Carolina Pardo-Tamayo, Tomás Herrera-Valenzuela

**Affiliations:** 1Escuela de Ciencias del Deporte y Salud, Facultad de Salud, Universidad Santo Tomás, Santiago 8370003, Chile; lifrancino@gmail.com (L.F.); bayronvm@outlook.com (B.V.); carolinapardo@santotomas.cl (C.P.-T.); 2Department of Physical Activity Science, Faculty of Education Sciences, Universidad Católica del Maule, Talca 3530000, Chile; pvaldes@ucm.cl; 3Carrera de Entrenador Deportivo, Escuela de Educación, Universidad Viña del Mar, Viña del Mar 2520000, Chile; 4Exercise and Rehabilitation Sciences Laboratory, School of Physical Therapy, Faculty of Rehabilitation Sciences, Universidad Andres Bello, Santiago 7591538, Chile; rodrigo.ramirez@unab.cl; 5Departamento de Ciencias del Deporte, Facultad de Ciencias de la Actividad Física, Universidad de Playa Ancha, Valparaíso 2340000, Chile; eduardo.baez@upla.cl; 6IRyS Group, Physical Education School, Pontificia Universidad Católica de Valparaíso, Valparaíso 2561427, Chile; alex.ojeda@pucv.cl; 7Escuela de Ciencias de la Actividad Física, el Deporte y la Salud, Facultad de Ciencias Médicas, Universidad de Santiago de Chile (USACH), Santiago 9170022, Chile; esteban.aedo@ind.cl; 8Laboratorio de Biomecánica Deportiva, Unidad de Ciencias Aplicadas al Deporte, Instituto Nacional de Deportes, Santiago 9170022, Chile

**Keywords:** athletic performance, high-intensity interval training, athletic performance, physical fitness, plyometric exercise, muscle strength, resistance training

## Abstract

The effect of multi-component training on specific performance is under-researched in wrestlers. The aim of this study was to determine the effect of six weeks of multi-component training on The Special Wrestling Fitness Test (SWFT) performances of wrestlers who were preparing for an international championship, and to, additionally, determine their inter-individual adaptive variability. The wrestlers (n = 13; 7 females; all international level) underwent technical-tactical and physical fitness training for the six weeks before the championship, 12 sessions per week (i.e., 36 h per week). Before and after the intervention the athletes were assessed with the SWFT, a wrestling-specific competitive performance test that includes measurements for throws, heart rate response to the SWFT, and the SWFT index. Significant pre–post intervention improvements were noted for throws (pre = 23.5 ± 2.9; post = 24.9 ± 3.6; *p* = 0.022) and SWFT_index_ (pre = 14.9 ± 2.2; post = 14.1 ± 2.2; *p* = 0.013. In conclusion, six weeks of multi-component training improved wrestling-specific competitive performances in highly-trained wrestlers, although with a meaningful inter-subject variability.

## 1. Introduction

Wrestling is characterized by high-intensity interval efforts (work:rest; 2.4:1) [1], and competitive performance level may be influenced by cardiorespiratory fitness, anaerobic performance (e.g., Wingate test), strength, and muscle power [2]. However, the physical fitness of athletes is commonly measured through tests that do not represent the specific characteristics of the sport [3], such as running and related ergometers. This situation has motivated the development of specific tests for wrestlers [3].

The Special Wrestling Fitness Test (SWFT), a specific test for wrestlers, has gained popularity [4,5,6,7,8]. The SWFT is the adaptation of another test originally designed for judo athletes (i.e., Special Judo Fitness Test) [9,10]. The SWFT involves measures of the autonomic nervous system related to wrestlers’ training levels [11], including heart rate immediately after the test (HR_final_), heart rate after one minute of rest after the test (HR_1min_), and the sum of HR_final_ and HR_1min_ (HR_sum_). The SWFT also measures the number of throws during the test, allowing the calculation of the SWFT_index_ (i.e., HR_sum_/throws). The SWFT is related to aerobic [4,5,6], anaerobic [4,6,7], and explosive strength performance [6], all proxies of competitive wrestling preparedness.

Judo athletes undergoing eight weeks of strength training improved the number of throws and the SWFT_index_, but without changes in HR_final_, HR_1min_, and HR_sum_ [12]. The authors of this study hypothesized that resistance training improved anaerobic alactic metabolism but not the aerobic component, since no change was observed in the heart rate response [12]. In another study, judo athletes undergoing 12 weeks of aerobic training improved in HR_final_, HR_1min_, and HR_sum_, but no changes were observed in the number of throws [13]. However, the effect of standard wrestling training (i.e., multi-component training) on the SWFT is under-researched. This precludes robust recommendations regarding the suitability of the SWFT to assess the preparation of competitive wrestlers, particularly before international events. Wrestlers typically perform different types of training in preparation for a match, for example, aerobic [13] and strength training [12]. Therefore, multi-component training of wrestlers is likely to improve throws and heart rate during SWFT.

Moreover, the competitive calendar of wrestlers usually obligates athletes to plan for four to eight weeks of training before competitions [14,15,16,17,18,19,20,21,22,23,24,25,26]. Therefore, further research is needed before robust claims regarding the suitability of the SWFT to assess international-level wrestlers’ competitive preparedness, and their anticipated inter-individual variability [16,17,18], after a short-term training period before competition.

Therefore, the aim of this study was to determine the effect of six weeks of training on the SWFT performance of wrestlers who were preparing for an international championship, and their inter-individual adaptive variability. Based on previous studies [12,13,20], we hypothesized that the training program would induce an increased performance in all variables of the SWFT (i.e., HR_final_, HR_1min_, HR_sum_, throws, and SWFT_index_).

## 2. Materials and Methods

The wrestlers underwent technical-tactical and physical fitness training for the six weeks before the South American Wrestling Championship, Santiago, Chile, 2019. The wrestlers trained one session in the morning (10:00 a.m.) and one afternoon session (5:00 p.m.), six days each week (Monday to Saturday). In addition, the athletes were evaluated before and after the 6 weeks of training through the SWFT. All tests and training were carried out at the Olympic Training Center of Chile. Figure 1 shows the experimental design.

### 2.1. Subjects

To calculate the required sample size, a freeware statistical software tool (G*Power; University of Düsseldorf, Düsseldorf, Germany) was used. The following variables were included in the a priori power analysis: number of tail: 2; effect size dz: 1.455, based on a previous study that investigated the effects of an aerobic training program on the SWFT performance of elite judo athletes [13]; alpha error: 0.05; correlation between groups: 0.5; desired power (1-ß error): 0.80. The results of the a priori power analysis indicated that a minimum of six participants would be needed per group to achieve statistical significance for SWFT performance. A greater number of participants was recruited considering potential dropouts due to injury, lack of time to attend measurement sessions, or related reasons.

A sample of 13 international-level wrestlers (7 females) from the Chilean national team participated in this study (males, age: 20.0 ± 4.6 years; body mass: 71.09 ± 6.91 kg; height: 1.66 ± 0.04 m; experience in competition: 5.0 ± 3.58 years; females, age: 20.6 ± 4.5 years; body mass: 61.27 ± 5.54 kg; height: 1.61 ± 0.05 m; experience in competition: 4.86 ± 3.34 years). Three athletes competed in Greco-Roman style, three in Freestyle, and seven in Female’s Wrestling. All were competitive wrestlers and met the following inclusion criteria: (i) more than one year of competitive experience; (ii) train six or more sessions per week. The wrestlers were free from any neuromuscular injuries and disorders and all the athletes in this study obtained a medal during the South American Wrestling Championship, Santiago, 2019. After being informed about the procedures and associated risks, all athletes provided written consent. This research was approved by the Institutional Ethics Committee (Universidad Santo Tomás, Code: 43.18).

### 2.2. Wrestling Training

Each typical training session (afternoon session) lasted approximately 180 min and consisted of technical and tactical exercises. The session began with a 20-min general warm-up, consisting of calisthenics, followed by throws and a low-intensity wrestling simulation. The central part of the training lasted 145 min and consisted of technical and tactical exercises to prepare the athlete for specific wrestling situations and high-intensity wrestling simulation. The 15-min cool-down included flexibility exercises (see Figure 1, Table 1 and Table 2).

### 2.3. Physical Fitness Training

The typical physical fitness training session consisted of endurance training (i.e., 30 min of running at moderate intensity), high-intensity interval training, short (i.e., 10 s all-out × 1-min rest × 6 reps), high-intensity interval training, long (i.e., 1 min at 100% maximal aerobic velocity × 3-min rest × 6 reps), sprint interval training (i.e., 6 s all-out × 3-min rest × 4 reps), maximal strength (i.e., 2 reps at 90% of one-repetition maximum (1RM) × 5 sets, with 3-min rest between sets), strength–endurance (i.e., 12 reps with 60% 1RM × 4 sets with 2-min rest between sets), muscle power (i.e., weightlifting, 4 reps with 85% 1RM × 4 sets, with 3-min rest) (see Figure 1, Table 1 and Table 2).

### 2.4. Special Wrestling Fitness Test

Before the test, the participants completed a 20-min warm-up, which included general and specific wrestling exercises that athletes normally perform during training. The test was performed on a wrestling mat (Dollamur FlexiRoll, Fort Worth, TX, USA) approved by United World Wrestling for international competitions [27]. The assessed athlete threw two other wrestlers during the test (who were 6 m apart from each other) as many times as possible in three sets (A: 15 s, B: 30 s, C: 30 s), with 10 s of rest between each set [4,5,6,7,8]. Greco-Roman wrestlers used an arm–neck throwing technique, while Freestyle and Women’s Wrestling used the fireman’s carry technique. This procedure aimed to determine the HR_final_, HR_1min_, and the total number of throws. The SWFT_index_ was calculated using the following equation: SWFT_index_ = HR_final_ + HR_1min_)/throws. The reliability of SWFT has been reported in well-trained male wrestlers through the intraclass correlation coefficient as throws (0.870), SWFT_index_ (0.867), HR_final_ (0.751), and HR_1min_ (0.703) [7].

### 2.5. Statistical Analysis

The distribution of the variables was examined using the Shapiro–Wilk test. The data are presented in mean and standard deviation. The 95% confidence interval and the percentage of change are presented for the differences. Sphericity was tested and confirmed using the Mauchly test. The Student *t* test (two moments) was used. As a measure of the effect size (ES), Cohen’s *d* was calculated following the classification proposed by Rhea for individuals training from 1–5 years (trivial < 0.35; small 0.35–0.80; moderate 0.80–1.50; large > 1.5) [28]. Following previous criteria [16,17,18,29], the non-responders (NRs) for each of the dependent variables were defined as athletes who were unable to demonstrate an increase or decrease (in favor of beneficial changes) that was greater than twice the typical error of measurement (TE) far from zero. On the contrary, the responders (Rs) for each of the dependent variables were defined as athletes who demonstrated a change beyond twice the TE, representative of a high probability (i.e., 12 to 1) of the observed response being a reflection of a true physiological adaptation, beyond what might be expected as a result of technical or biological variability. The TE was as follows: SWFT_index_, 0.73 (arbitrary unit); HR_final_, 5.27 (beats per minute [bpm]); HR_1-min_, 9.67 (bpm); HR_sum_, 11.81 (bpm); and throws, 1.34 (n).

## 3. Results

The results for the dependent variables, namely, throws, SWFT_index_, HR_final_, HR_1min_, and HR_sum_, are summarized in Figure 2 and Figure 3, and Table 3.

Briefly, significant pre–post intervention improvements were noted for throws (pre = 24 ± 3; post = 25 ± 4; *p* = 0.022, t = 2.64, df = 12, Figure 2A) and SWFT_index_ (pre = 14.91 ± 2.16; post = 14.07 ± 2.22; *p* = 0.013, t = 2.93, df = 12, Figure 2B), but there were no effects for HR_final_ (pre = 186 ± 7 bpm; post = 183 ± 4 bpm; *p* = 0.153, t = 1.52, df = 12), HR_1min_ (pre = 159 ± 20 bpm; post = 159 ± 13 bpm; *p* = 0.878, t = 0.20, df = 12) and HR_sum_ (pre = 345 ± 25; post = 342 ± 14; *p* = 0.331, t = 0.52, df = 12). Among the five dependent variables, the pre–post relative changes after training intervention were between 0.6% to −6.5% (Table 3), the ES between 0.04 to 0.47 (Table 3), the non-responders 77% to 100% (Figure 3) and involved 0 to 3 responders (Table 3).

## 4. Discussion

We hypothesized that the training program would increase performance in all variables of the SWFT (i.e., HR_final_, HR_1min_, HR_sum_, throws, and SWFT_index_), but we partially corroborate our hypothesis, because the number of throws and the SWFT index improved, but we did not find significant improvements in HR_final_, HR_1min_, and HR_sum_.

The number of throws improved as well as the SWFT_index_, probably due to improvements in aerobic and anaerobic performances [6]. Indeed, the number of throws during SWFT and the SWFT_index_ were correlated with explosive strength measured through countermovement jump with and without arms, squat jump, and reactivity strength index, as well as with maximal oxygen uptake (VO_2max_) [4,6] and with anaerobic performance assessment through the Wingate test [4,7].

Our study found no significant improvements in HR_final_, HR_1min_, and HR_sum_. Such findings replicated those found regarding judo athletes after 8 weeks of either linear or undulating periodized strength training, where throws and SJFT_index_ improved, without changes in HR_final_, HR_1min_, and HR_sum_, probably due to improvements in anaerobic alactic metabolism [12]_._ On the other hand, in judo athletes, 12 weeks of aerobic training induced improvements in the SJFT, specifically in HR_final_, HR_1min_, and HR_sum_, but no changes were observed in the number of throws [13]. Therefore, differences between our results and those from previous studies may be partially related to different training configurations. For example, in our study the participants completed 6 weeks of training, combining cardiorespiratory fitness and muscle power-oriented training exercises. Such a time frame allowed positive adaptations in the aerobic and anaerobic performance of wrestlers [24], both variables related to the number of throws during the SWFT [4,6].

Regarding the analysis of the inter-individual responses, our results showed 0% of responders for HR_sum_, 15% for HR_final_, and 8% for HR_1min_, while we observed 23% of responders for the number of throws and the SWFT_index_. According to Walsh et al. [30], the inter-individual response is a combination of the following: (i) individual response to perseverative exercise training (subject-training interaction), (ii) day-to-day biological variation and technical error (random variation), and (iii) physiological response associated with behavioral/maturational changes, not attributable to exercise (e.g., within-person variability) [30]. This is the first study to analyze inter-individual response in wrestling athletes, observing a low percentage of responders, probably due to the high level of training of the athletes. In other Olympic combat sports, where the inter-individual response to 4 weeks of HIIT has been analyzed, between 0% and 87.5% of responders were found for aerobic performance, and between 0% and 50% for jump performance [16,17,18,26], which are variables related to SWFT [4,6]. In the present study, the high number of non-responders to training probably reflected the difficulty to further improve the athletic performance of athletes already at the elite level, particularly after a relatively reduced number of training weeks. Moreover, the athletes were in preparation for the most important event of their competitive year, meaning that before enrolment in this study, they were already near their peak performance before major competition, with little room for further improvement. Alternatively, the high number of training hours and, thus, training density, may have induced cumulative fatigue, precluding visualization of athletic performance improvements after a brief period (i.e., 48 h) of rest from training. Future studies might need to confirm this hypothesis. Although this hypothesis is unlikely, since the athletes obtained positive results during the competition.

A limitation of the present study is the lack of a control group. A control group is usually difficult in elite-level athletes, as modifications in the training program of the athletes are usually not well received by the athletes or coaches. Moreover, a control group was precluded due to the limited sample of available participants (elite-level athletes). Indeed, competitive teams are usually composed of a small number of athletes of both genders. Therefore, studies with small samples of males and females are common [16,18,26]. On the other hand, studies that analyze the effects of training in a national team for an international tournament are scarce and have high ecological validity. In the present study, the national wrestling team were prepared for the South American Championship, Santiago. 2019, where all the athletes in the sample obtained a gold, silver, or bronze medal.

Lastly, coaches can improve the number of throws and the SWFT_index_ in the last six weeks before a competition. This is a relevant fact, since the SJFT is a test that discriminates between athletes of different competitive levels, and its improvement can increase the chances of success in the competition [31].

## 5. Conclusions

In conclusion, six weeks of multi-component training improved wrestling-specific competitive performance in highly-trained wrestlers, although with a meaningful inter-subject variability. Therefore, coaches can use a multi-component training program when approaching relevant competitions or when a short training period is available before competitions.

## Figures and Tables

**Figure 1 ijerph-19-09325-f001:**
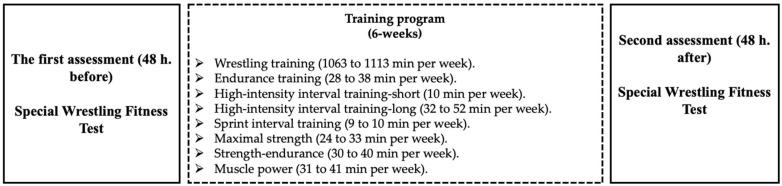
Experimental design.

**Figure 2 ijerph-19-09325-f002:**
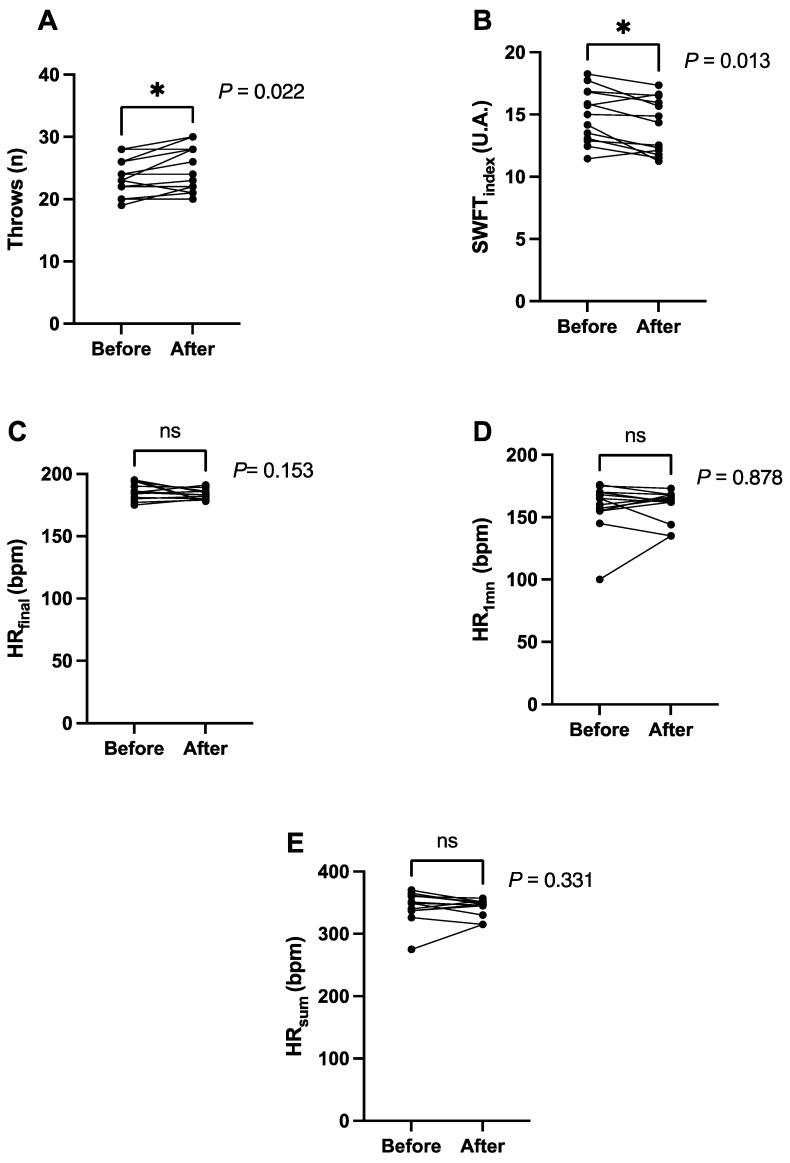
Effect of 6 weeks of multi-component training for (**A**) Throws, (**B**): Special Wrestling Fitness Test index (SWFT_index_)_,_ (**C**) HR_final_ (heart rate at the end of the SWFT), (**D**) HR_1min_ (heart rate at 1-min post SWFT), (**E**) HR_sum_ (sum of HR_final_ and HR_1min_), *: Statistically significant differences (*p* ≤ 0.05), ns: Not statistically significant.

**Figure 3 ijerph-19-09325-f003:**
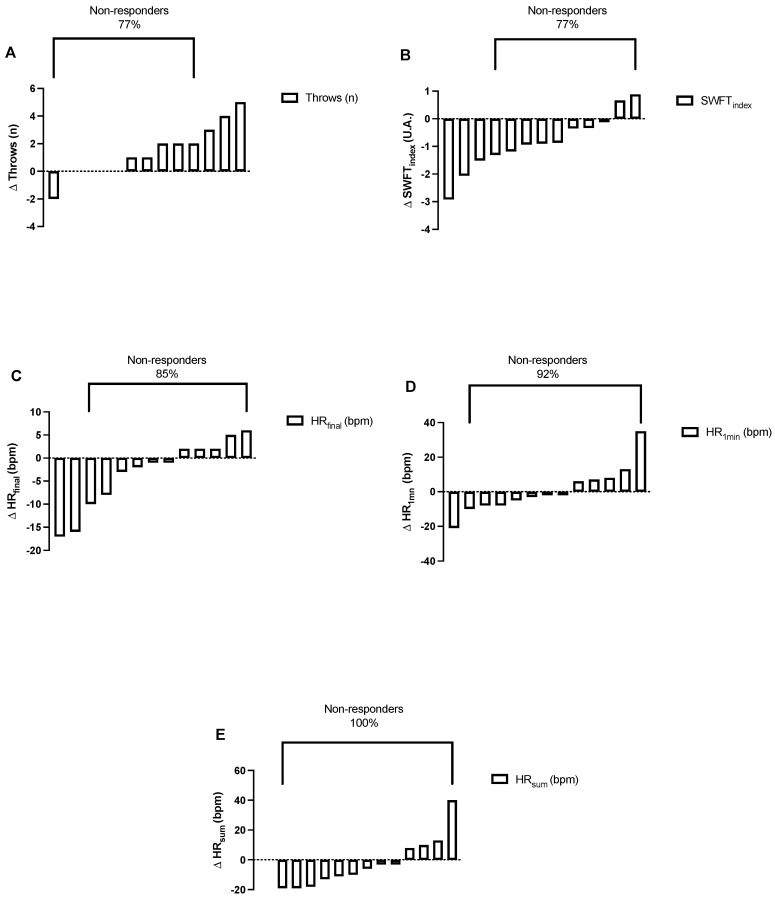
Individual pre–post change [Δ], after 6 weeks of multi-component training, for (**A**) Throws, (**B**): Special Wrestling Fitness Test index (SWFT_index_)_,_ (**C**) HR_final_ (heart rate at the end of the SWFT), (**D**) HR_1min_ (heart rate at 1-min post SWFT), (**E**) HR_sum_ (sum of HR_final_ and HR_1min_).

**Table 1 ijerph-19-09325-t001:** Volume (min) of training during each week.

Training	Week 1	Week 2	Week 3	Week 4	Week 5	Week 6
Endurance training (min)	38	36	34	32	30	28
HIIT-short (min)	10	10	10	10	10	10
HIIT-long (min)	52	48	44	40	36	32
SIT (min)	10	10	10	9	9	9
Muscle power (min)	41	40	37	36	33	31
Strength-endurance (min)	40	38	36	34	31	30
Maximal strength (min)	33	31	30	28	25	24
Wrestling training (min)	1113	1102	1093	1082	1074	1063
Total	1337	1315	1294	1271	1248	1227

HIIT-short: High-Intensity Interval Training-short; HIIT-long: High-Intensity Interval Training-long; SIT: Sprint Interval Training.

**Table 2 ijerph-19-09325-t002:** Example of the distribution of weekly training time.

Training	Monday	Tuesday	Wednesday	Thursday	Friday	Saturday	Sunday
Endurance training (min)	---	---	---	---	---	38	---
HIIT-short (min)	---	---	---	---	10	---	---
HIIT-long (min)	---	26	---	26	---	---	---
SIT (min)	10	---	---	---	---	---	---
Muscle power (min)	20	---	21	---	---	---	---
Strength-endurance (min)	15	---	15	---	10	---	---
Maximal Strength (min)	---	---	---	---	33	---	---
Wrestling training (min)	180	185	190	180	180	198	---

HIIT-short: High-Intensity Interval Training-short; HIIT-long: High-Intensity Interval Training-long; SIT: Sprint Interval training.

**Table 3 ijerph-19-09325-t003:** Group pre–post change and responders after 6 weeks of multi-component training.

Variables	% Change (SD)	Rs, n (%)	Effect Size	95% CI for the Mean
Throws (n)	5.1 (7.3)	3 (23)	0.47 (small)	0.24 to 2.53
SWFT_index_ (AU)	−6.5 (8.3)	3 (23)	−0.39 (small)	−1.47 to −0.22
HR_final_ (bpm)	−1.8 (4.1)	2 (15)	−0.46 (small)	−7.66 to 1.35
HR_1min_ (bpm)	0.6 (9.6)	1 (8)	0.04 (trivial)	−7.50 to 9.04
HR_sum_(bpm)	0.6 (5.1)	0 (0)	−0.10 (trivial)	−12.48 to 7.71

n = number; AU: arbitrary unit; bpm: beats per minute; SD: standard deviation; Rs: responders; CI: confidence interval.

## Data Availability

The datasets generated during and/or analyzed during the current research are available from the Corresponding author on reasonable request.

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
