# Peer review of "Effect of a Six Week In-Season Training Program on Wrestling-Specific Competitive Performance"

_ijerph, 2022, doi:10.3390/ijerph19159325_

Round 1

Reviewer 1 Report

See comments in attached document

Author Response

Reviewer 1

In my opinion, the abstract is generally correct. The abstract contains the following information: aim, sample, the method used, results and conclusion. I believe that the summary could be improved by adding at the beginning (before the aim) a sentence containing the following information in summary form: summary of the background and the problems detected.

R: We appreciate the reviewer's suggestions and have adjusted as follows: “The effect of multi-component training on the specific performance is under-researched in wrestlers”.

1.- Introduction: The authors explain the performance factors that affect wrestlers. They also explain how we can measure these performance factors using tests specific to wrestlers: Special Wrestling Fitness Test (SWFT). The authors elaborate on the scientific background of the SWFT. And they point out the aspects that we can measure with this test. The authors detect the problems of previous studies. The authors justify the need for their study. I consider that the development of the introduction and the justification of the study are correct.

R: Thank you very much for your comments

  1. Method: I consider that the materials and methods used are correct. The authors clearly explain the design of the study. The authors present the sample size calculations. They sufficiently describe the characteristics of the participants and the inclusion criteria. The participants provided written consent. The study was approved by an ethics committee. All procedures and statistical analyses are fully explained. This allows the study to be replicated.

R: Thank you very much for your comments

  1. Results: I consider that the results are clear and respond to the aim of the study. The results are sufficiently reflected in Figures 3 and 4, Table 3 and the text.

R: Thank you very much for your comments

  1. Discussion: I think the discussion is well thought out. All results achieved are discussed with appropriate scientific papers. Nevertheless, I would like to make one suggestion. Suggestion At the end of the introduction, the authors state the following hypothesis: “Based on previous studies [18,25,26] we hypothesized that the training program would induce an increased performance in all variables of the SWFT (i.e., HRfinal, HR1min, HRsum, throws, and SWFTindex)”. Based on this hypothesis, in the discussion, the authors should explicitly confirm or reject this hypothesis. Therefore, I suggest that the authors modify the first two paragraphs of the discussion by alluding to this aspect.

R: We appreciate the reviewer's suggestions and have adjusted as follows: “We hypothesized that the training program would increase performance in all variables of the SWFT (i.e., HRfinal, HR1min, HRsum, throws, and SWFTindex), but we partially corroborate our hypothesis, because the number of throws and the SWFT index improved, but we did not find significant improvements in HRfinal, HR1min, and HRsum”.

  1. Conclusion: I believe that the conclusions are correct.

R: Thank you very much for your comments

References I do not understand reference 27.

R: We appreciate the reviewer's suggestions and have adjusted as follows: “United World Wrestling Licensed mats Available online: https://uww.org/governance/licensed-mats (accessed on May 22, 2022)”.

Reviewer 2 Report

Dear Authors,

The submitted manuscript was not developed in accordance with the standards of scientific articles. The scientific level of the manuscript is insufficient. The weakest parts of the manuscript: Introduction, Discussion, Conclusions. Conducted comparative analysis is very poor. There is no in-depth analysis of the research results.

Detailed comments:

1.     Key words: The use of some words seems unjustified e.g. martial arts (the manuscript is only about wrestling), human physical conditioning, sports (too wide a range).

2.     Introduction: very poor. Authors should briefly introduce the problem, particularly emphasizing the level of knowledge about the problem at the beginning of the investigation. The Authors mainly enumerate articles (see: line 70-71; line 74).

3.     Methods

- Figure 2: unnecessary.

- Table 2: ʺExamle of the distribution of weekly training timeʺ. Please explain why this information was provided?

- the part 2.3.:the descriptionis is too general.

- the part 2.4.: the title should not be an acronym (SWFT).

4.     Results

Research results need to be described, not just presented in tables or figures (compare: https://www.mdpi.com/journal/ijerph/instructions#preparation)

5.     Discussion.

Discussion should include interpretation of study findings, and results considered in the context of results in other studies reported in the literature. Do not repeat in detail data or other material from the Introduction or the Results part. Authors should discuss the results and how they can be interpreted in perspective of previous studies and of the working hypotheses.

6.     Conclusions: It was written unjustified conclusion: ʺThis data confirm the sensibility of a the SWFTʺ? It was not be the aim of the research (compare: p. 2 line 72-73, the aim of the study). It was not describe in which scientidic way it was estmated. References:

- the selection of references was insufficient, for 30 articles only about 8 concern wrestling;

- different font;

- No 9 and No 10: other surname of the first author (incorrect spelling);

- No 2, 12, 28: there is error in the name (an abbreviation) of the journal;

- No 27: ʺUnited World Wrestling No Titleʺ. Such a literature item should not be included in the list.

Author Response

Reviewer 2

The submitted manuscript was not developed in accordance with the standards of scientific articles. The scientific level of the manuscript is insufficient. The weakest parts of the manuscript: Introduction, Discussion, Conclusions. Conducted comparative analysis is very poor. There is no in-depth analysis of the research results.

Detailed comments:

1.- Key words: The use of some words seems unjustified e.g. martial arts (the manuscript is only about wrestling), human physical conditioning, sports (too wide a range).

R: We appreciate the reviewer's suggestions and have adjusted as follows: “Keywords: martial arts; athletic performance; high-intensity interval training; athletic performance; physical fitness; plyometric exercise; muscle strength; resistance training.

2.- Introduction: very poor. Authors should briefly introduce the problem, particularly emphasizing the level of knowledge about the problem at the beginning of the investigation. The Authors mainly enumerate articles (see: line 70-71; line 74).

R: We appreciate the reviewer's suggestions and have adjusted as follows: “Judo athletes undergoing eight weeks of strength training improved the number of throws and the SWFTindex, but without changes in HRfinal, HR1min, and HRsum[12]. While judo athletes undergoing 12 weeks of aerobic training improved in HRfinal, HR1min, and HRsum, but no changes were observed in the number of throws[13]”. However, the effect of standard wrestling training (i.e., multi-component training) on the SWFT is under-researched. This precludes robust recommendations regarding the suitability of the SWFT to assess competitive wrestler’s preparation, particularly before international events.

  1. Methods

- Figure 2: unnecessary.

R: Removed.

- Table 2: ʺExamle of the distribution of weekly training timeʺ. Please explain why this information was provided?

R: We think it is relevant to present Table 2 because it shows the distribution of training volume within one week. This allows other authors and trainers to replicate the program.

- the part 2.3.:the descriptionis is too general.

R: At this point, our objective is to present an example of the training content in each session.

- the part 2.4.: the title should not be an acronym (SWFT).

R: Adjusted.

  1. Results

Research results need to be described, not just presented in tables or figures (compare: https://www.mdpi.com/journal/ijerph/instructions#preparation)

R: We appreciate the reviewer's suggestions and have adjusted as follows: Briefly, significant pre-post intervention improvements were noted for throws (pre = 24 ± 3; post = 25 ± 4, Figure 2 A) and SWFTindex (pre = 14.91 ± 2.16; post = 14.07 ± 2.22, Figure 2 B), but no there were no effects (P > 0.05) for HRfinal (pre = 186 ± 7 bpm; post = 183 ± 4 bpm), HR1min (pre = 159 ± 20 bpm; post = 159 ± 13 bpm) and HRsum (pre = 345 ± 25; post = 342 ± 14). Among the five dependent variables, the pre-post relative changes after training intervention were between 0.6% to -6.5% (Table 3), the ES between 0.04 to 0.47 (Table 3), the non-responders 77% to 100% (Figure 3) and involved 0 to 3 responders (Table 3).

  1. Discussion.

Discussion should include interpretation of study findings, and results considered in the context of results in other studies reported in the literature. Do not repeat in detail data or other material from the Introduction or the Results part. Authors should discuss the results and how they can be interpreted in perspective of previous studies and of the working hypotheses

R: Adjusted.

  1. Conclusions: It was written unjustified conclusion: ʺThis data confirm the sensibility of a the SWFTʺ? It was not be the aim of the research (compare: p. 2 line 72-73, the aim of the study). It was not describe in which scientidic way it was estmated. 

R: We appreciate the reviewer's suggestions and have adjusted as follows: “In conclusion, six weeks of multicomponent training improves wrestling-specific competitive performance in highly-trained wrestlers, although with a meaningful inter-subject variability. This data confirm the sensibility of the SWFT to short periods of intense preparation for competition in highly trained wrestlers”

References:

- the selection of references was insufficient, for 30 articles only about 8 concern wrestling;

R: We reference all articles on the existing SWFT. In addition, we used various articles on SJFT with judo athletes for analysis.

- different font;

- No 9 and No 10: other surname of the first author (incorrect spelling);

R: Adjusted.

- No 2, 12, 28: there is error in the name (an abbreviation) of the journal;

R: Adjusted.

- No 27: ÊºUnited World Wrestling No Titleʺ. Such a literature item should not be included in the list.

R: Adjusted.

Reviewer 3 Report

Thanks to the authors for an interesting study and manuscript.

The article needs corrections:

Abstract:

The introduction should clearly state what problem this research is solving. It is necessary to indicate the main references of the study object. It is also necessary to explain how the community of scientists and practitioners can use the information obtained in the study. In the presented version, the abstract is overloaded with methodological information, which is irrelevant in the abstract. The content of the Introduction is controversial. Does not match manuscript titles.

Keywords:

I suggest reducing to 5-6 keywords that are directly related to the research and avoiding informational noise. For example, the term martial arts is irrelevant in this work.

Materials and Methods

The authors incorrectly describe/present Physical fitness training content (Figure 1. Experimental design).

It is necessary to accurately classify the physical loads applied during training. I suggest following this source: https://us.humankinetics.com/blogs/excerpt/understanding-energy-systems-training    Periodization of Strength Training for Sports - 4th edition by Bompa.

It is not clear whether the authors are looking for a suitable fitness assessment tool (SWFT) or evaluating the effectiveness of a 6-week training cycle?

Discussion and conclusion

It must discuss and reveal the value of the research data and compare it with the research data of other authors.

Conclusions have to summarize the results of the research and identify their benefits for other researchers and practitioners (reveal the practical value of the work).

The manuscript requires substantial revisions.

Author Response

Reviewer 3

Abstract:

The introduction should clearly state what problem this research is solving. It is necessary to indicate the main references of the study object. It is also necessary to explain how the community of scientists and practitioners can use the information obtained in the study. In the presented version, the abstract is overloaded with methodological information, which is irrelevant in the abstract. The content of the Introduction is controversial. Does not match manuscript titles.

R: We appreciate the reviewer's suggestions. We incorporated the changes that were in our capacity.

Keywords:

I suggest reducing to 5-6 keywords that are directly related to the research and avoiding informational noise. For example, the term martial arts is irrelevant in this work.

R: Adjusted.

Materials and Methods

The authors incorrectly describe/present Physical fitness training content (Figure 1. Experimental design).

R: Adjusted.

It is necessary to accurately classify the physical loads applied during training. I suggest following this source: https://us.humankinetics.com/blogs/excerpt/understanding-energy-systems-training    Periodization of Strength Training for Sports - 4th edition by Bompa.

It is not clear whether the authors are looking for a suitable fitness assessment tool (SWFT) or evaluating the effectiveness of a 6-week training cycle?

R: Dear reviewer, we respectfully believe that we have detailed the training program as much as possible.

The objective of this study was to determine the effect of 6-weeks of training on the SWFT performance in wrestlers

Discussion and conclusion

It must discuss and reveal the value of the research data and compare it with the research data of other authors.

Conclusions have to summarize the results of the research and identify their benefits for other researchers and practitioners (reveal the practical value of the work).

R: We appreciate the reviewer's suggestions. We incorporated the changes that were in our capacity.

We hope that the new version of the manuscript will be satisfactory for publication.

Round 2

Reviewer 2 Report

Dear Authors,

In my opinion the submitted manuscript is still not developed in accordance with the standards of hifh-performance scientific articles. Your additions are too superficial. Your manuscript is not adequate for journal articles with a high Impact Factor. You particularly poorly responded to points 2, 4, 5, 6 of my comments from Round 1. The weakest parts of the manuscript: Introduction, Discussion, Conclusions. 1-2 sentence supplements are definitely insufficient.

Author Response

Dear Reviewer,

Thank you very much for your comments. We have made the modifications that were within our reach. We know that our manuscript has some limitations, specifically not having a control group and the sample size. However, these limitations were mentioned in the manuscript; on the other hand, studies with these characteristics have high ecological validity.

Reviewer 3 Report

Thanks to the authors for the exciting research. 

I think the manuscript can be published in the last form. 

Sincerely.

Author Response

Dear Reviewer,

Thank you very much for your comments. We think that the manuscript was improved thanks to your comments.  
